# GDF11 Regulates PC12 Neural Stem Cells via ALK5-Dependent PI3K-Akt Signaling Pathway

**DOI:** 10.3390/ijms232012279

**Published:** 2022-10-14

**Authors:** Zongkui Wang, Peng Jiang, Fengjuan Liu, Xi Du, Li Ma, Shengliang Ye, Haijun Cao, Pan Sun, Na Su, Fangzhao Lin, Rong Zhang, Changqing Li

**Affiliations:** 1Institute of Blood Transfusion, Chinese Academy of Medical Sciences & Peking Union Medical College, Chengdu 610052, China; 2Sichuan Blood Safety and Blood Substitute International Science and Technology Cooperation Base, Chengdu 610052, China

**Keywords:** growth differentiation factor 11, neural stem cells, proliferation, apoptosis, migration, PI3K-Akt pathway

## Abstract

Growth differentiation factor 11 (GDF11), belonging to the transforming factor-β superfamily, regulates anterior-posterior patterning and inhibits neurogenesis during embryonic development. However, recent studies recognized GDF11 as a rejuvenating (or anti-ageing) factor to reverse age-related cardiac hypertrophy, repair injured skeletal muscle, promote cognitive function, etc. The effects of GDF11 are contradictory and the mechanism of action is still not well clarified. The objective of the present study was to investigate effects of GDF11 on PC12 neural stem cells in vitro and to reveal the underlying mechanism. We systematically assessed the effects of GDF11 on the life activities of PC12 cells. GDF11 significantly suppressed cell proliferation and migration, promoted differentiation and apoptosis, and arrested cell cycle at G2/M phase. Both TMT-based proteomic analysis and phospho-antibody microarray revealed PI3K-Akt pathway was enriched when treated with GDF11. Inhibition of ALK5 or PI3K obviously attenuated the effects of GDF11 on PC12 neural stem cells, which exerted that GDF11 regulated neural stem cells through ALK5-dependent PI3K-Akt signaling pathway. In summary, these results demonstrated GDF11 could be a negative regulator for neurogenesis via ALK5 activating PI3K-Akt pathway when it directly acted on neural stem cells.

## 1. Introduction

Around the world, advances in medicine technology and public health measures and improvements in living standards have greatly prolonged human life span [1]. With the number of aged persons precipitously increasing, more and more people suffer from age-related cognitive decline [2,3]. Age-related cognitive decline is a common degenerative disease of the central nervous system, of which Alzheimer’s disease (AD) is the most well-known form. In the United States, an estimated 6.2 million people over 65 years old are living with AD, which makes AD presently the 6th leading cause of death [4]. There are approximately 10 million AD patients in China, and AD is the fifth leading cause of death among urban and rural residents [5,6]. Age-related cognitive decline, with the features of long disease duration, slow progress, and complex etiology, remains incurable, and there are even a few drugs that can prevent or delay its progress. AD that brings heavy burdens on the patients, their families and society, has become an important social issue. A looming question is to develop new strategies for treating, preventing, or delaying age-related cognitive decline.

Recent studies suggested that the blood of young mice could fight against ageing of old mice. They found young blood were with effects of reversing age-related cardiac hypertrophy, repairing injured skeletal muscle, increasing dendritic spine density, improving cognitive function, etc. [7,8,9]. The rejuvenation of young blood has attracted growing attentions in the past few years. However, direct transfusion of young blood raised a crucial ethical issue that the poor young people would be reduced to sell their blood for the rejuvenation of a small “elite” of people [10,11]. Therefore, it is crucial to seek young blood-associated factors, i.e., rejuvenating factors. In 2013, Loffredo et al. [7] identified growth differentiation factor 11 (GDF11) as a pro-youthful factor from young blood to reverse age-related cardiac hypertrophy. Subsequently, other studies revealed GDF11 could restore injured skeletal muscle, enhance neurogenesis, improve cognitive function, and so on [8,9,12].

GDF11, also known as bone morphogenetic protein 11 (BMP11), is a member of transforming growth factor β (TGF-β) superfamily [13,14]. GDF11, highly homologous with myostatin (also known as GDF8), participates in modulating anterior-posterior patterning, and is considered as a negative regulator of muscle growth and neurogenesis [15,16]. Those are apparently contradictory with the positive rejuvenative effects of GDF11. Likewise, several other studies questioned the positive effect of GDF11, and they implied that GDF11 was not with the effect of anti-ageing, but a potential blockade target for treating age-related diseases. Egerman et al. [17] displayed GDF11 inhibited skeletal muscle regeneration. Smith and colleagues [18] also suggested GDF11 did not rescue aging-related pathological hypertrophy. Furthermore, GDF11 was considered to inhibit bone formation, decrease bone mass, and negatively regulate neurogenesis [19,20,21]. Taken together, until now, it is incongruous whether GDF11 is a rejuvenating factor or a pro-ageing factor. Additionally, the effects and mechanism of action of GDF11 on neurogenesis are still not yet fully elucidated.

Therefore, the aim of the present study was to investigate the potential effects of GDF11 on PC12 neural stem cells in vitro and to reveal the underlying mechanism of action. Firstly, we studied the effects of GDF11 on proliferation, migration, differentiation, apoptosis, and cell cycle of PC12 neural stem cells. Thereafter, proteomic and signal pathway microarray analysis were performed to seek the key proteins and signal pathways involved in GDF11 regulating PC12 cells. Finally, the underlying mechanism was revealed by inhibiting the key signal pathway.

## 2. Results

### 2.1. GDF11 Mitigated Cell Viability and Proliferation

We first analyzed the effects of GDF11 on cell viability of PC12 cells by enhanced Cell Counting Kit-8 (CCK-8) assay. PC12 cells were cultured using starved medium with various concentrations of GDF11 (0, 12.5, 25, 50 and 100 ng/mL) for a 72 h period. GDF11 with 25 and 100 ng/mL slightly yet significantly increased (less than 10%, *p* < 0.05) cell viability after 24 h cultivation. Meanwhile, other concentrations of GDF11 showed no effects on cell viability. When treated for 48 h, GDF11 did not affect the cell viability. Only 100 ng/mL GDF11 a small magnitude reduction on cell viability (less than 10%, *p* < 0.05) when assessed at 72 h (Figure 1A).

Results of xCELLigence RTCA indicated both GDF11- and vehicle-treated cells significantly proliferated after 120 h cultivation (Figure 1B). At approximately the 50th h, compared to the control group, 12.5 ng/mL GDF11 showed no effect on PC12 cell proliferation, whereas 25, 50, and 100 ng/mL GDF11 significantly suppressed cell growth (*p* < 0.01). From the 70th h, the addition of GDF11 resulted in obvious suppression of proliferation (*p* < 0.05) at any of the time points compared with the control. Of note, there was a slight dose-dependent suppressing effect of GDF11 on cell proliferation.

We performed the assay of cell population doubling times to investigate the long-term effect of GDF11 on the proliferation of neural stem cells. PC12 cells, with or without GDF11 treatment, all displayed significant proliferative effects for the five-passage duration (Figure 1C). GDF11 showed no substantial effect on PC12 proliferation at P1 and P2, though 50 ng/mL GDF11 marginally decreased proliferation (*p* < 0.05). From P3, higher concentrations of GDF11 (50 and 100 ng/mL) significantly inhibited PC12 cell proliferation (*p* < 0.05), whereas low concentrations of GDF11 did still not affect the cell growth yet. At P5, the accumulative population doubling times were 29.47 (control), 26.93 (12.5 ng/mL), 26.02 (25 ng/mL; *p* < 0.05 compared with control), 22.92 (50 ng/mL; *p* < 0.05 compared with control), and 22.04 (100 ng/mL; *p* < 0.05 compared with control), respectively. That meant 100 ng/mL GDF11 resulted in the lowest accumulative population doubling times (~25.2% lower than control; *p* < 0.05) after a five-passage cultivation.

Taken together, GDF11 showed limited effects on the viability of PC12 cells during a 48 h treatment, whereas higher concentrations of GDF11 significantly attenuated cell proliferation in a long-term cultivation.

### 2.2. GDF11 Promoted Differentiation of PC12 Cells

As shown in Figure 1D,E, western bolt results revealed GDF11 slightly decreased protein levels of the neural progenitor cell marker, nestin. Conversely, protein levels of βIII-tubulin (neuronal biomarker) and GFAP (astrocytic biomarker) of the GDF11-treated groups were increased when compared to the control. Similar with protein levels, the mRNA expression of nestin declined and that of βIII-tubulin and GFAP increased when exposed to GDF11 (Figure 1F, Appendix A). These results suggested that GDF11 promoted neural stem cells to neuronal and astrocytic differentiation.

### 2.3. GDF11 Stimulated Apoptosis of PC12 Cells

The flow cytometery results of Annexin V-FITC/PI dual staining illuminated that the apoptotic and necrotic cells following GDF11 treatment were significantly increased (Figure 2A,B). Only 14.4% of PC12 cell were apoptotic in the control group, whereas 17.8%, 23.5% (*p* < 0.05), 27.2% (*p* < 0.05) and 30.5% (*p* < 0.05) cells displayed apoptosis when treated with 12.5, 25, 50 and 100 ng/mL GDF11 for 72 h, respectively. Compared to the control, exposure to 100 ng/mL GDF11 stimulated approximately double apoptosis cells. Furthermore, GDF11 showed a slight dose-dependent effect to induce apoptosis.

### 2.4. GDF11 Arrested Cell Cycle of PC12 Cells

As shown in Figure 2C,D, GDF11 treatment did not change the proportion of PC12 cells in S phase of cell cycle. Compared to the control (66.63%), the percent of G0/G1 cells treated with GDF11 was 62.86% (12.5 ng/mL; *p* < 0.05), 60.28% (25 ng/mL; *p* < 0.05), 56.67% (50 ng/mL; *p* < 0.01), and 56.36% (100 ng/mL; *p* < 0.05), respectively. For the cells in G2/M phase, GDF11 obviously increased the proportion from 14.21% (control) to 17.19% (12.5 ng/mL; *p* < 0.05), 17.96% (25 ng/mL; *p* < 0.05), 21.83% (50 ng/mL; *p* < 0.01), and 25.07% (100 ng/mL; *p* < 0.05), respectively. The results indicated that GDF11 significantly arrested PC12 cell cycle in G2/M phase, which could partly explain the effects of GDF11 on inhibiting proliferation and promoting apoptosis of PC12 cells.

We also evaluated transcriptional levels of the key proteins of cell cycle (i.e., cyclin D1, cyclin D2, cyclin A, and cyclin B1). The results of qRT−PCR revealed that GDF11 hardly affected the mRNA levels of cyclin D1 and cyclin A, whereas GDF11 significantly decreased transcription of cyclin D2 and cyclin B1 (Figure 3A). These might, in part, be the underlying mechanism of GDF11-induced cell cycle G2/M arrest.

### 2.5. GDF11 Attenuated PC12 Cell Migration

In the results of wound healing/migration assay, the wound area decreased as PC12 cell migration progressed over time (Figure 3B,C). After 12 h of cultivation, low concentrations of GDF11 played no roles in wound closure, with the wound area was 72.6% (control), 73.5% (12.5 ng/mL GDF11), and 73.6% (25 ng/mL GDF11), respectively. However, higher concentrations of GDF11 significantly suppressed wound closure of PC12 cells, with the wound area 82.3% for 50 ng/mL GDF11 treatment (*p* < 0.05) and 80.6% for 100 ng/mL GDF11 treatment (*p* < 0.05), respectively. As the cultivation time prolonged to 36 h, GDF11 greatly attenuated migration, with the wound area of 12.3% (control), 28.1% (12.5 ng/mL GDF11; *p* < 0.01), 36.4% (25 ng/mL GDF11; *p* < 0.001), 34.8% (50 ng/mL GDF11; *p* < 0.001), and 40.3% (100 ng/mL GDF11; *p* < 0.01), respectively. All of these results indicated GDF11 significantly attenuated migration of PC12cells.

### 2.6. TMT–Based Protein Identification and Quantification

The SDS-PAGE results of the control and 100 ng/mL GDF11-treated group showed good consistency of the samples in each group (Appendix A). The results of the MS data analysis (including the distributions of peptide length and protein mass) indicated that the sample preparation procedures matched the requirement and were reliable (Appendix A). A total of 5 209 proteins were quantifiable (Table 1), and the distribution information of the identified proteins (including isoelectric point, molecular weight, and peptide coverage) were shown in Appendix A. Results showed that 165 proteins were differentially expressed, of which 90 were up-regulated and 65 were down-regulated in GDF11-treated group when compared with control (Figure 4A,B and Appendix A).

### 2.7. Annotation Analysis of the Differentially Expressed Proteins

GO enrichment analysis was performed to analyze the biological significance of the aforementioned 165 differentially expressed proteins (DEPs). The detailed GO annotation of the DEPs was shown in Figure 4D and Appendix A. Briefly, the analysis of biological process revealed the DEPs were mainly enriched in phosphatidylinositol phosphorylation, cell cycle arrest, phosphatidylinositol-mediated signaling, glycerophospholipid metabolic process, and so on. And for the molecular function analysis, the DEPs were mainly involved in transcription cofactor activity, transcription co-activator activity, and protein kinase C activity.

To further understand the functional differences of the dysregulated proteins, KEGG pathway analysis was implemented. The top 20 enriched KEGG pathways were displayed in Figure 4C, including the pathways of Ras signaling, PI3K-Akt signaling, HIF-1 signaling, phosphatidylinositol signaling system, and sphingolipid metabolism. From the GO and KEGG enrichment analysis, we found the DEPs were closely involved in phosphatidylinositol signaling system, including phosphatidylinositol 3-kinase (PI3K)-Akt pathway, which implied PI3K-Akt pathway was participated in the modulating effects of GDF11 on PC12 cells.

Protein-protein interaction (PPI) network analysis via STRING database and Cytoscape software obviously revealed that at least three crucial cross-talk signaling clusters were displayed in the complex PPI network, and the key nodes were Pik3ca (coding for the p110α catalytic subunit of PI3K), Trp53bp1, and Cep290 (Figure 5A). After simplifying the PPI network, Pik3ca was the visible key node (Figure 5B), which further suggested PI3K-Akt was closely associated with GDF11 effects on PC12 cells.

### 2.8. PI3K-Akt Pathway Was Enriched by CSP100 PLUS Microarray

Detailed information of the 304 unphosphorylated proteins and phosphorylated sites in the CSP100 PLUS microarray was displayed in Appendix A. Treatment with 100 ng/mL GDF11 altered 42 unphosphorylated protein expressions, of which 19 down-regulated and 23 up-regulated (Appendix A). Compared to the control, the phosphorylation levels of 25 proteins were up-(21/25) or down-regulated (4/25) in the GDF11-treated group (Figure 6A). The significant up-regulation of phosphorylation of PI3K, GSK3, p70S6K, HSP27, Smad2, and Smad3 were the key proteins in PI3K-Akt and TGFβ-Smad2/3 signal pathways.

We performed an integrated analysis of the 16 signaling pathway of the microarray to explore the core signal-net (Appendix A and Figure 6B), and it illuminated PI3K and Akt2 were the key proteins involved in the core signal-net. Further analysis of pathway cross-talk revealed PI3K-Akt pathway was closely related with cell cycle, apoptosis and AMPK pathway after GDF11 treatment (Appendix A). And PI3K-Akt was the fundamental signaling pathway involved in the effects of GDF11 on neural stem cells (Figure 6C).

### 2.9. GDF11 Activated PI3K-Akt Signal Pathway

Because both TMT-based proteomic analysis and CSP100 PLUS microarray indicated PI3K-Akt was significantly altered when PC12 cells were treated with GDF11, we investigated PI3K-Akt pathway to reveal the underlying mechanism of the effects of GDF11 on PC12 cells. The results of western blot displayed 100 ng/mL GDF11 treatment significantly increased the phosphorylation levels of both PI3K and Akt, when compared with the control (Figure 7A,B, Appendix A). These results showed GDF11 activated PI3K-Akt signaling pathway in neural stem cells and were consistent with the results of the TMT-based proteomic analysis and CSP100 PLUS microarray.

### 2.10. GDF11 Affected Neural Stem Cells through ALK5-Dependent PI3K-Akt Pathway

Given that members of TGF-β family bind to TGF-β receptors to form complex which phosphorylates the downstream proteins to regulate cellular behaviors. We then measured whether GDF11-induced PI3K-Akt pathway depended on its binding to the TGF-β type 1 receptor (TGFβRI or ALK5), as it was previously suggested [22]. In the present study, we first inhibited ALK5 with SB431542, a specific ALK5 inhibitors. We found SB431542 showed no cytotoxicity on PC12 cells (Figure 7C). Results of western blot displayed that inhibiting ALK5 by SB431542 strongly diminished phosphorylation levels of Smad2/3, PI3K and Akt (Figure 7D,E, Appendix A). Furthermore, blocking ALK5 abolished the effects of GDF11 on PC12 cells. Evaluating in a detailed manner, the effects of GDF11-induced suppression of proliferation and migration (Figure 8A–C), promotion of differentiation and apoptosis (Figure 8D,E and Figure 7F–H), and arrest of cell cycle (Figure 8F) were reversed by inhibiting ALK5 with SB431542. These results demonstrated that GDF11 triggered downstream signaling pathways via an ALK5-dependent manner to regulate the life activities of neural stem cells.

Subsequently, we explored the effects of GDF11-induced PI3K-Akt pathway on neural stem cells by blocking PI3K with a PI3K inhibitor LY294002. Treatment with LY294002 decreased the GDF11-induced Akt phosphorylation, without affecting the phosphorylation levels of Smad2/3 (Figure 7D,E, Appendix A). These indicated that co-treatment with GDF11 and LY294002 inhibited the PI3K-Akt pathway, while it activated TGF-β/Smad2/3 pathway. Moreover, LY294002 attenuated GDF11-mediated inhibition of proliferation and migration (Figure 8A–C), increase of differentiation and apoptosis (Figure 8D–E and Figure 7F–H), and cell cycle arrest (Figure 8F), which suggested PI3K-Akt signaling pathway was crucial for the effects of GDF11 on neural stem cells.

Taken together, GDF11 regulated cellular behaviors of neural stem cells via ALK5-dependent PI3K-Akt signaling pathway.

## 3. Discussion

In this study, we demonstrated, in vitro, GDF11 suppressed cell proliferation and migration, promoted differentiation and apoptosis, and induced cell cycle arrest in the G2/M phase. And the effects of GDF11 were diminished when blocking ALK5 by SB431542 or inhibiting PI3K by LY294002. GDF11 regulated the life activities of neural stem cells via ALK5-dependent activation of downstream PI3K-Akt signaling pathway.

At present, our understanding on the function of GDF11 in ageing and neurogenesis was still quite limited and controversial. GDF11, as a member of TGF-β superfamily, modulates anterior-posterior patterning during mammalian embryonic development [15,23]. In the initial stages, GDF11 was recognized as a negative regulator of neurogenesis [16], which was consistent with other studies [21,24], and our results showing that GDF11 inhibited PC12 cell proliferation and promoted differentiation. However, GDF11 attracted much attention for its rejuvenating effect on neurogenesis in recent years [8,25,26], which was contradictory with the negative regulation on neurogenesis. The recent study by Mayweather et al. [21] explained this phenomenon. They revealed endogenous GDF11 in brain suppressed hippocampal neurogenesis through directly contacting with neural stem cells. In contrast, the circulatory GDF11 in blood, which was recognized as a pro-youthful factor to increases hippocampal neurogenesis, was perhaps to play indirect roles on neural stem cells without crossing the blood-brain barrier. In our present study, we used the in vitro model to investigate the effects of GDF11 on neural stem cells. GDF11 in the medium acted directly on PC12 cells, just like the manner of endogenous GDF11 directly acting on hippocampal neural stem cells. Therefore, our findings of the effects of GDF11 on suppression of proliferation and promotion of differentiation are reliable. It appears that GDF11 is a “pro-ageing factor” when it directly acts on neural stem cells, whereas GDF11 is a “rejuvenating factor” when it indirectly acts on neural stem cells. However, it is still unclear whether the effects of endogenous GDF11 and circulatory GDF11 on other cells (e.g., skeletal muscle cells and cardiomyocytes) are the same, and this requires further investigation. It also raises an interesting suggestion for further studies that GDF11 or other proteins may have different (or even totally opposite) effects on the same cell, depending on the direct or indirect action.

Furthermore, we found GDF11 not only promoted apoptosis but also suppressed migration and cell cycle of neural stem cells. In line with our results, both Wu et al. [16] and Williams et al. [24] showed that GDF11 arrested cell cycle of neural stem cells. And Williams et al. [24] also supported our results that GDF11 diminished migration of neural stem cells in vitro. However, Katsimpardi et al. [8] found the circulatory GDF11 in blood promote migration of neural cells, which was in contrast with our in vitro results. It should be noted that circulatory GDF11 showed indirect effects on neural stem cells whereas GDF11 directly acted on neural stem cells in our present study. Therefore, the results of Katsimpardi et al. [8] cannot be simply compared with our results.

As a member of TGF-β superfamily, GDF11 binds to TGF-β receptors to activate the downstream signaling pathways, including the classical TGF-β/Smad and non-Samd pathways [13,27]. In the present study, both proteomic and CSP100 PLUS microarray results displayed PI3K-Akt signaling pathway was significantly enriched after GDF11 treatment. Using western blot, we revealed GDF11 activated both the classical TGF-β/Smad2/3 and non-Smad PI3K-Akt pathways. Numerous studies have demonstrated GDF11 could activate classical TGF-β/Smad (including TGF-β/Smad2/3 and TGF-β/Smad1/5/8) signal pathway to regulate cell behaviors in multiple cell types, such as osteoblasts, human umbilical vein endothelial cells, neural stem cells, mice retinal microvascular endothelial cells, and bone marrow mesenchymal stem cells [14,20,28,29]. Similarly, GDF11 also activated non-Smad (e.g., AMPK, MAPK, mTOR, PI3K) pathways in a wide range of cell types, such as cardiomyocytes, neural stem cells, human hepatocellular carcinoma cells, primary hepatocytes, and mesenchymal stem cells [14,30,31,32,33]. Through the ALK5 inhibitor SB431542, we further revealed GDF11 depended on ALK5 to activate the downstream TGF-β/Smad2/3 and PI3K-Akt pathways to regulate cell behaviors of PC12 cells. Thereafter, when inhibited PI3K by the inhibitor LY294002, results implied that PI3K-Akt (not TGF-β/Smad2/3) signaling pathway was crucial for GDF11 to regulate neural stem cells. Similar to our results, Zhang et al. [33] found GDF11 enhanced mesenchymal stem cells viability, mobility, and angiogenic paracrine functions through ALK5-dependent PI3K-Akt pathway, and Frohlich et al. [22] showed that the effects of GDF11 on lipid accumulation were also via ALK5-dependent PI3K-Akt pathway. However, we still do not know whether there is crosstalk between the TGF-β/Smad2/3 signaling pathway and the PI3K-Akt signaling pathway, and it needs further studying to reveal the underlying mechanism.

Given that cell signaling interaction is not a simple linear regulation, and it is impossible that GDF11 regulates neural stem cells solely through PI3k-Akt pathway, other factors or pathways may also be involved in GDF11 regulatory network. Seong and Kang [34] revealed that miR-1260b directly targeted the 3′UTR of GDF11 to mediate GDF11-Smad-dependent signaling to regulate proliferation of vascular smooth muscle cell. circUCK2 inhibited miR-125b-5p activity to increase GDF11 expression, resulting in amelioration of neuronal injury in mice, which suggested that the circUCK2/miR-125b-5p/GDF11 axis was an essential signaling pathway involved in ischemia stroke [35]. GDF11 significantly attenuated the severity of skin inflammation by suppressing NF-κB signaling pathway in both imiquimod-induced mice model and IL-23-induced mice model [36]. As described above, GDF11 regulates cell behaviors through complex signaling regulatory networks. More studies should be performed to further refine the GDF11-mediated regulatory network.

## 4. Materials and Methods

### 4.1. Cell Culture

PC12 cells were purchased from zqxzbio (Shanghai, China) and cultured with complete medium (DMEM supplemented with 5% (*v*/*v*), fetal bovine serum (FBS; Procell Life Science & Technology Co., Ltd., Wuhan, China), 10% (*v*/*v*) horse serum (HS; Biological Industries, Israel), 100 U/mL penicillin and 100 µg/mL streptomycin) at 37 °C, and 5% CO_2_ in a humidified atmosphere. After reaching approximately 80% confluence, PC12 cells were trypsinized and sub-cultured. For following experiments, the medium was changed to starved medium (DMEM supplemented with 0.5% FBS, 1% HS, 100 U/mL penicillin and 100 µg/mL streptomycin) after cultivation overnight. After 6 h of starvation, different concentrations of GDF11 (12.5, 25, 50 and 100 ng/mL; R & D systems, Minneapolis, MN, USA) were added, respectively. For signal pathway inhibition, after 6 h starvation, PC12 cells were pretreated with ALK5 inhibitor SB431542 (Beyotime Biotechnology, Shanghai, China) or PI3K inhibitor LY294002 (Beyotime Biotechnology, China) for 4 h before GDF11 was added for a further 5–72 h.

### 4.2. Cell Viability/Proliferation Assay

Cell viability was evaluated by CCK-8 (Beyotime Biotechnology, China) at time points of 24, 48, and 72 h. PC12 cells were cultured in 96-well plates at a density of 3000 cells/well in 200 μL complete medium overnight. Thereafter, complete medium was replaced with starved medium for 6 h to synchronize growth. Various concentrations of GDF11 were supplemented. According to the manufacturer’s instructions, 20 µL of CCK-8 agent was added to each well 2 h before the termination of the experiment. The absorbance was detected at 450 nm using a SpectraMax M2^e^ (Molecular Devices, San Jose, CA, USA). Then cell viability was calculated by comparing the optical density (OD) values of GDF11-treated with untreated cells.

Proliferation of PC12 cells was also measured by a real time cell analyzer (xCELLigence RTCA TP, ACEA Biosciences, San Diego, CA, USA), which is an impedance-based technology that can be used for real-time monitoring and noninvasively quantifying cell proliferation. PC12 cells were cultured in an E-16 plate at a density of 2000 cells/well, and then the plate was inserted into the xCELLigence station at 37 °C with 5% CO_2_ for 120 h. Data were collected every 10 min, and then the proliferation curve was drawn.

Cell population doubling times (PDTs) were performed to assay cell proliferation as well. PC12 cells were seeded in a 96-well plate at a density of 3000 cells/well. PC12 cells were passaged every 48 h, and cells collected at the end of each passage were counted using an automatic image cytometer (Cellometer K2; Nexcelom Bioscience, Lawrence, MA, USA). Cell population doubling times was calculated using the following formula: PDTs = Pn/P0, where P0 and Pn represent the cell number at the initial seeding passage (0) and at a specific passage (n), respectively.

### 4.3. Cell Apoptosis Assay

For cell apoptosis assay, a cell apoptosis assay kit (Beyotime Biotechnology, China) based on Annexin V-FITC and propidium iodide (PI) dual staining was used on a FACS flow cytometry (Celesta^TM^; BD, San Jose, CA, USA). The cell populations were discriminated according to their position of quadrants as we previously reported [14]: live cells (Annexin V−/PI−), early/primary apoptotic cells (Annexin V+/PI−), late/secondary apoptotic cells (Annexin V+/PI+), and necrotic cells (Annexin V−/PI+).

### 4.4. Wound Healing/Migration Assay

For wound healing/migration assay, Culture-Insert 4 Well in µ-Dish 35 mm (Ibidi GMBH, Germany) was used. PC12 cells were employed to each well of the Culture-Insert at a density of 5000 cells/well with complete medium. When reaching 70–80% confluence, the medium was replaced by starved medium for another 6 h. Then the Culture-Insert was removed to generate cell-free gaps (i.e., wound) with a width of 500 μm, and various concentrations of GDF11 diluted with starved medium (0 ng/mL, 12.5 ng/mL, 25 ng/mL, 50 ng/mL, and 100 ng/mL, respectively) were added. Images were captured at 0 h, 12 h and, 36 h with an inverted microscope to measure the wound closure. The percent of wound closure of various treatments at different time points were calculated with Wimasis software (Onimagin Technologies, Córdoba, Spain).

### 4.5. qRT-PCR Analysis

Total RNA was extracted from the cultured cells using TRIZOL reagent (Invitrogen, Waltham, MA, USA) according to the manufacturer’s standard procedure. Total RNA (1 µg) was used for reverse transcription in a final volume of 20 µL. Subsequently, qRT-PCR was performed using the Quantitect SYBR Green PCR kit (Qiagen, Germantown, MD, USA) with an ABI StepOnePlus Real-Time PCR System (Applied Biosystems, Waltham, MA, USA). The relative expressions of target genes were calculated using the 2^−ΔΔCt^ method by normalizing with the housekeeping gene (GAPDH or β-actin) expression and presented as fold changes relative to control. The PCR primers were synthesized by Beijing Genomics Institute (Shenzhen, China), and the sequences of primers are displayed in Appendix A.

### 4.6. Western Blot Analysis and Validation

Cells were lysed by RIPA buffer (Beyotime Biotechnology, China) supplemented with 1× protease inhibitor cocktail and 1× phosphatase inhibitor cocktail. Total protein was measured by using a BCA protein assay kit (Beyotime Biotechnology, China). Then, equal amount proteins (25 µg) of each sample were denatured with NuPAGE 4× LDS loading buffer (Invitrogen, USA), and were separated by SDS-PAGE (NuPAGE 4–12% Bis-Tris gel; Invitrogen, USA). After transferred to PVDF membranes, proteins were incubated with the following rabbit antibodies at 4 °C overnight: anti-nestin (Cell Signal Technology, Danvers, MA, USA), anti-βIII-tubulin (Cell Signal Technology, USA), anti-GFAP (Cell Signal Technology, USA), and anti-GAPDH (Cell Signal Technology, USA), anti-BCL-2 (Cell Signal Technology, USA), anti-Bax (Cell Signal Technology, USA), anti-Cleaved caspase-3 (Cell Signal Technology, USA), anti-Smad2/3 (Cell Signal Technology, USA), anti-PI3K p85 (Cell Signal Technology, USA), anti-Akt (Cell Signal Technology, USA), anti-p-Smad2/3 (Cell Signal Technology, USA), anti-p-PI3K p85 (Cell Signal Technology, USA), anti-p-Akt (Cell Signal Technology, USA), and β-actin (Cell Signal Technology, USA), respectively. Subsequently, the PVDF membranes were incubated with horseradish peroxidase-labeled goat anti-rabbit IgG (ZSGB-BIO, Beijing, China) at room temperature for 2 h, and then were visualized by ECL reagents (Foregene Co., Ltd., Chengdu, China).

### 4.7. Sample Preparation, TMT-Labeling, and LC-MS/MS

PC12 cells were cultured in 6-well plates with complete medium overnight. Then the medium was replaced by starved medium for another 6 h. The group of C1, C2, and C3 was treated with 100 ng/mL GDF11 supplemented with starved medium, while the controls of A1, A2, and A3 were treated with starved medium. The cells were lysed with lysis buffer (8 M urea, 1% protease inhibitor). Protein concentration was determined by a BCA kit (Beyotime Biotechnology, China), and the consistency of each group was detected by SDS-PAGE.

The detailed methodology of protein digestion, TMT-labeling and LC-MS/MS were performed as we previously described [3,37,38]. Briefly, at the beginning, proteins of each sample (100 µg) were reduced, alkylated, and digested according to the protocols. Subsequently, the digested peptides were labeled with the TMT label reagents according to the instructions of 6-plex TMT labeling kit (Thermo Fisher Scientific, Waltham, MA, USA). Sample labeling was as follows: group A1: 126, group A2: 127, group A3: 128, group C1: 129, group C2: 130, and group C3: 131.

The labeled samples were subsequently fractionated into 60 fractions by high pH reverse-phase high-performance liquid chromatography (HPLC). Then, the 60 fractions were combined into 18 fractions, and vacuum-dried. The dried samples were subsequently reconstituted and detected by LC-MS/MS on a Q Exactive Plus mass spectrometer (Thermo Scientific) that was coupled to an EASY-nLC 1000 (Thermo Fisher Scientific, USA) ultra-performance liquid chromatography (UPLC) system.

### 4.8. MS/MS Data Analysis and Bioinformatics Analysis

The raw MS/MS data were searched using the MASCOT engine (version 2.2) embedded into Proteome Discoverer 1.4 software with a false discovery rate (FDR) < 1% at protein, peptide, and PSM levels and minimum score for peptides > 40. The peptide mass tolerance for precursors was set as 20 ppm in the first search and 5 ppm in the main search, and the 0.02 Da was for fragment ions. The requirements of all identified proteins were with at least 2 peptides with ≥1 unique peptide, and only the proteins at *p* < 0.05 were considered to be accurately quantified. For comparisons between GDF11treat groups and controls, a protein with a fold change of > ±1.2 (*p* < 0.05) was presented as a DEP. DEPs were analyzed according to GO terms for biological process, cellular component, and molecular function using the UniProt-GOA database (http://www.ebi.ac.uk/GOA/ accessed on 12 June 2020). Domain functional descriptions of the DEPs were predicted by using InterProScan soft (http://www.ebi.ac.uk/interpro/ accessed on 12 June 2020). Pathway enrichment analysis was assessed using KEGG pathway database (http://www.genome.jp/kegg/ accessed on 13 June 2020). Protein−protein interaction (PPI) networks were analyzed by the STRING database (http://string-db.org accessed on 12 June 2020), and visualized by Cytoscape software (version 3.6.0), respectively.

### 4.9. Profiles of Key Signaling Pathway by an Antibody Array

The key signaling pathways induced by GDF11 in neural stem cells were profiled using a phosphorylation-profiling antibody microarray CSP100 PLUS (Full Moon Microsystems, Charlotte, MI, USA) performed by Wayen Biotechnology (China) according to the protocols. A total of 304 unphosphorylated proteins and phosphorylated sites in 16 signaling pathways were analyzed. The obtained data were first normalized by housekeeping protein, and then the ratio of protein expression and phosphorylation changes were calculated by comparing them to the control. Differentially expressed proteins or phosphorylation sites were set with a fold change > ±1.2. The differentially phosphorylated proteins were analyzed using Ingenuity Pathway Analysis (IPA; QIAGEN, Hilden, Germany) to reveal the core signal-net and pathways.

### 4.10. Statistical Analysis

The statistical analysis was performed using GraphPad Prism 8.0.2 and the data were presented as the mean ± standard error of the mean (SEM) otherwise specified. Hierarchical clustering heat map of the DEPs was conducted by using HemI 2.0 (https://hemi.biocuckoo.org/ accessed on 4 March 2022). Multi-group comparisons were conducted by one-way ANOVA followed by Tukey’s post hoc test. Paired analysis of control and GDF11-treated groups was performed using two-tailed paired Student’s *t* test. In all analyses, *p* < 0.05 was considered statistically significant.

## 5. Conclusions

Our in vitro study found that GDF11 showed substantially effects on PC12 neural stem cells, which was suppression of proliferation and migration, promotion of apoptosis and differentiation, and arrest of cell cycle. Further proteomic and microarray analysis revealed PI3K-Akt pathway was significantly enriched when PC12 cells were treated with GDF11. By blocking ALK5 or PI3K, we demonstrated that GDF11 regulated the life activities of neural stem cells via binding ALK5 to activate PI3K-Akt signaling pathway. These results implied that GDF11 could be a potential target for pharmacologic blockade, not a rejuvenative factor, when GDF11 directly acted on neural stem cells, and could develop new therapeutic strategy for neurogenesis.

## Figures and Tables

**Figure 1 ijms-23-12279-f001:**
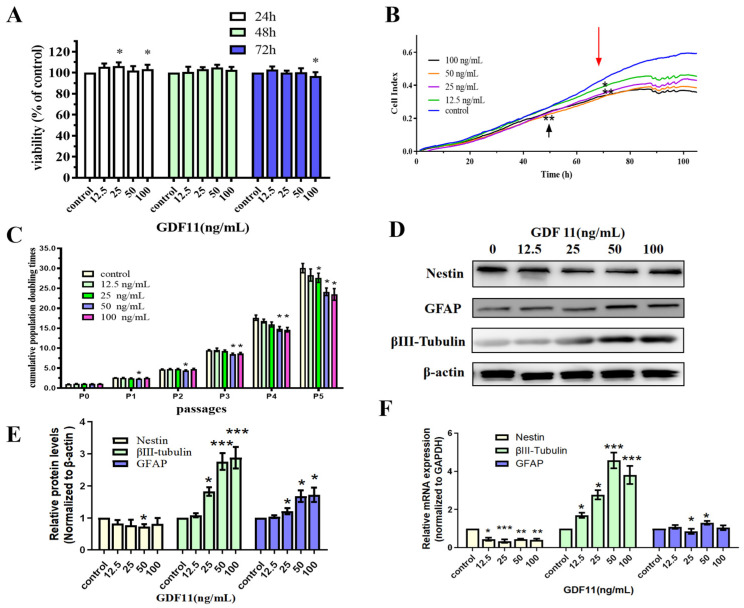
Effects of GDF11 on PC12 cells. (**A**) The viability of PC12 cells treated with various concentrations of GDF11 or vehicle for 24 h, 48 h, or 72 h was measured by CCK-8 assay. * *p* < 0.05. (**B**) Cell viability was real-time monitored by a real time cell analyzer (xCELLigence RTCA TP) for a 120 h-period. * *p* < 0.05, ** *p* < 0.01. (**C**) Accumulative population doubling times (PDTs) of PC12 cells treated with different GDF11 concentrations for a total period of 5 passages. * *p* < 0.05. (**D**) Protein levels of nestin, βIII-tubulin and GFAP were altered when PC12 cells were treated with various concentrations of GDF11 (0, 12.5, 25, 50 and 100 ng/mL) for 72 h. (**E**) Quantitative analyses of protein expressions of (**D**) in relation to β-actin expression. * *p* < 0.05, *** *p* < 0.001. (**F**) Transcriptional levels of nestin, βIII-tubulin and GFAP when PC12 cells were incubated with or without GDF11 for 5 h. * *p* < 0.05, ** *p* < 0.01, *** *p* < 0.001.

**Figure 2 ijms-23-12279-f002:**
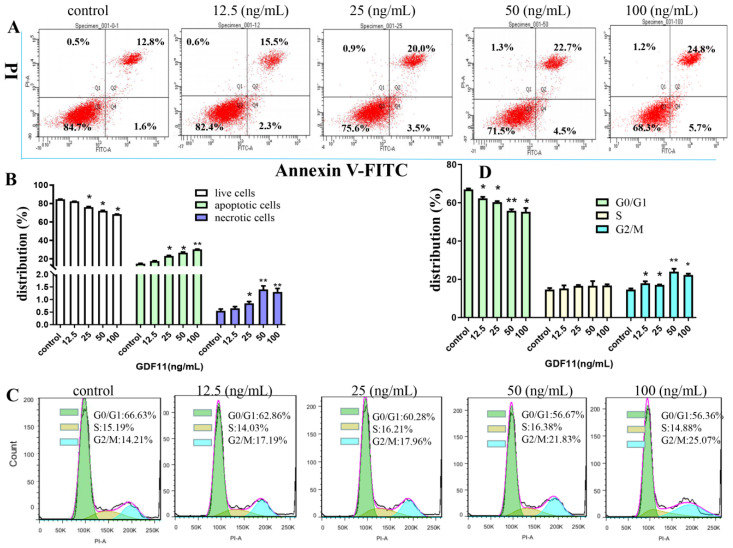
Effects of GDF11 on apoptosis and cell cycle of PC12 cells. (**A**) GDF11 promoted apoptosis in PC12 neural stem cells. PC12 cells were treated with various concentrations of GDF11 (0, 12.5, 25, 50, and 100 ng/mL) for 72 h, and then were assessed on a flow cytometry by using Annexin V−FITC and PI dual staining: Lower left quadrant; live cells (Annexin V−/PI−); Lower right quadrant: early/primary apoptotic cells (Annexin V+/PI−); Upper right quadrant: late/secondary apoptotic cells (Annexin V+/PI+); Upper left quadrant: necrotic cells (Annexin V−/PI+). (**B**) Quantitative analyses of apoptosis as in (**A**). (**C**) GDF11 induced cell cycle arrest at G2/M phase. PC12 cells were incubated with GDF11 (0, 12.5, 25, 50 and 100 ng/mL) or vehicle for 72 h and analyzed by Flow cytometry. (**D**) Histogram of cell cycle distribution as in (**C**). * *p* < 0.05, ** *p* < 0.01.

**Figure 3 ijms-23-12279-f003:**
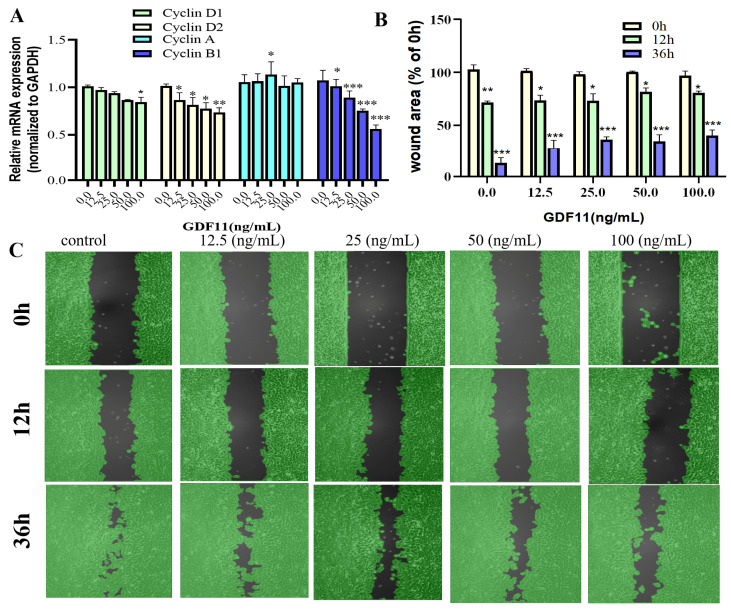
Effects of GDF11 on migration and cell cycle-related proteins. (**A**) The mRNA levels of cell cycle-related proteins (Cyclin D1, D2, A and B1) when PC12 cells were treated with GDF11 or vehicle (control) for 5 h. (**B**) Quantitative analysis of migration as in (**C**). (**C**) Wound healing/migration was monitored over time by an Ibidi Culture-Insert. Black in each graph represented as the wound area. * *p* < 0.05, ** *p* < 0.01, *** *p* < 0.001.

**Figure 4 ijms-23-12279-f004:**
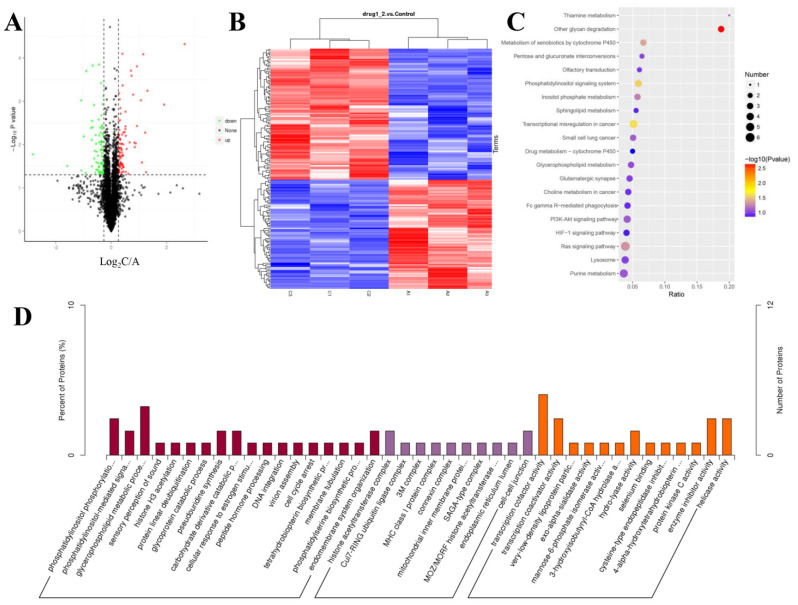
Characteristics of the identified proteins in PC12 cells with or without GDF11-treated. (**A**) Volcano plot of the 165 differentially expressed proteins in GDF11-treated group when compared to GDF11-untreated group. A protein with a fold change > ±1.2 and *p* < 0.05 was regarded as a differentially expressed (up-regulated or down-regulated) protein, which was colored in red or green, respectively. The identified proteins with no significant changes were in black. (**B**) Hierarchical clustering heat map of the 165 differentially expressed proteins. A1, A2, and A3 represented the 3 replicates of the GDF11-untreated group, and C1, C2, and C3 represented the 3 replicates of GDF11-treated PC12 cells. (**C**) The top 20 enrichment KEGG pathway of the differentially expressed proteins. (**D**) GO enrichment analysis of the DEPs in three categories, i.e., molecular function, cellular component, and biological process.

**Figure 5 ijms-23-12279-f005:**
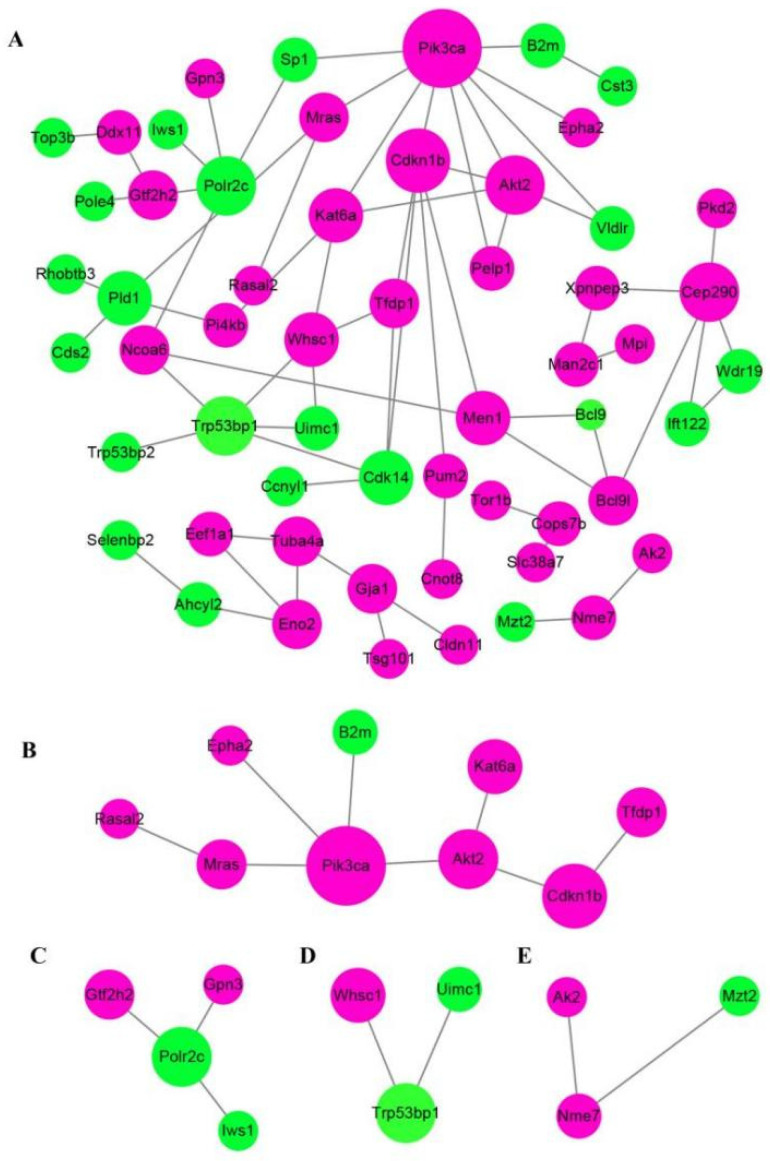
Analysis of protein−protein interaction network of the differentially expressed proteins. Functional interactions of the 165 DEPs were annotated by STRING database and visualized by Cytoscape software. (**A**) The total PPI network of the 165 DEPS. (**B**–**E**) were the simplified core PPI networks.

**Figure 6 ijms-23-12279-f006:**
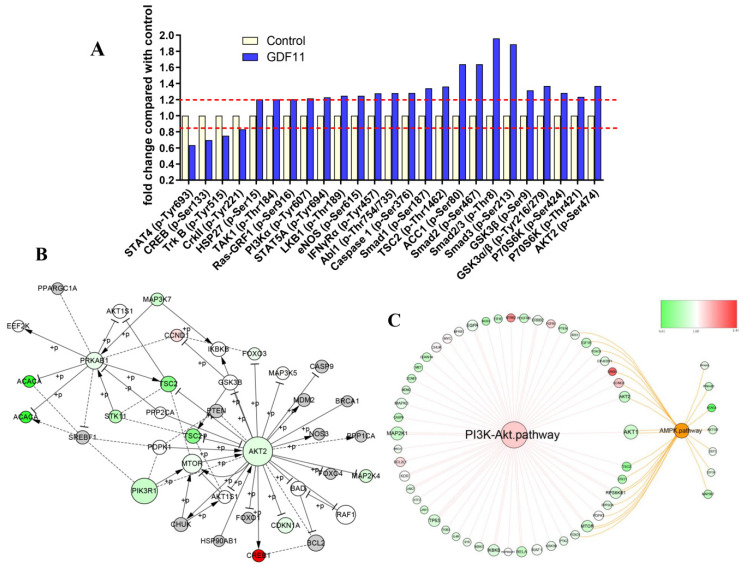
PI3K-Akt pathway was enriched by CSP100 PLUS Microarray. (**A**) Histogram showed the relative fold changes of the differentially expressed proteins when treated with GDF11. The dotted lines were set at ±1.2 and considered as the differentially expressed proteins. (**B**) The core signal-net were conducted by Ingenuity pathway analysis (IPA). (**C**) The key pathway cross-talk analysis was performed by IPA.

**Figure 7 ijms-23-12279-f007:**
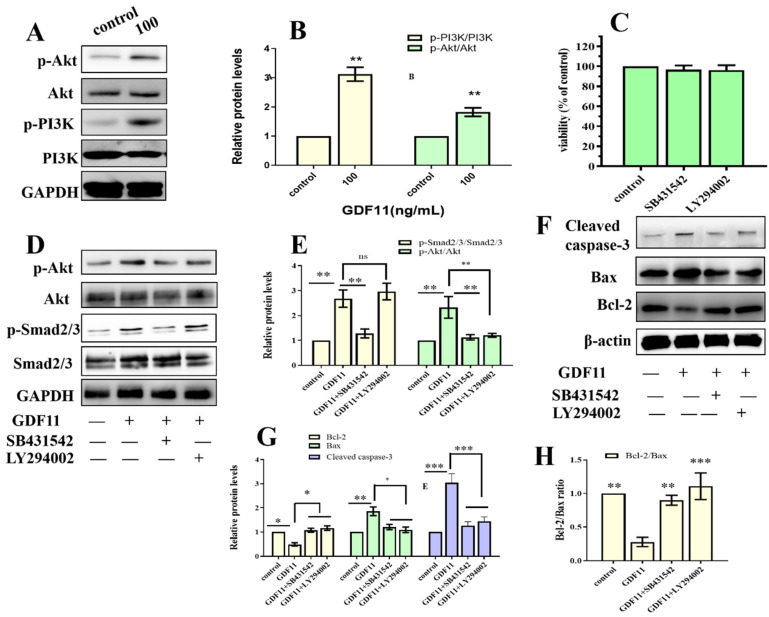
GDF11 activated PI3K-Akt pathway via ALK5. (**A**) GDF11 activated PI3K-Akt pathway. PC12 cells were treated or untreated with 100 ng/mL for 24 h and analyzed by western blot. (**B**) Histogram of (**A**). (**C**) Cell proliferation assessment in PC12 cells incubated with SB431542 (20 μM) or LY294002 (1 μM). (**D**) Protein and phosphorylation levels of AKT, p-AKT, Smad2/3 and p-SMAD2/3 in PC12 cells treated or untreated wtihGDF11 (100 ng/mL), SB431542 (20 μM) or LY294002 (1 μM) for 5 h. (**E**) Quantitative analysis of (**D**). (**F**) The apoptosis-related proteins were detected by western blot. (**G**,**H**) were Quantitative analysis of (**F**). * *p* < 0.05, ** *p* < 0.01, *** *p* < 0.001.

**Figure 8 ijms-23-12279-f008:**
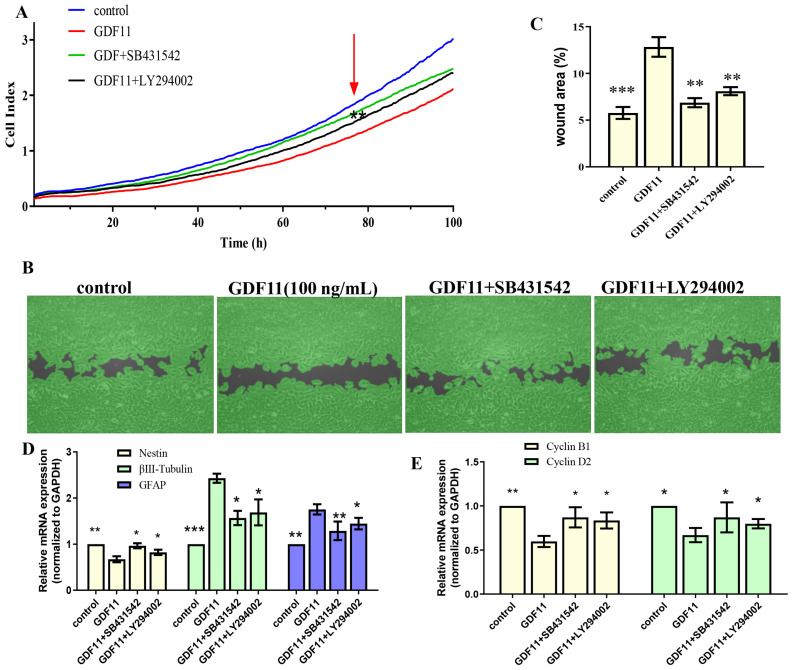
The effects of GDF11 on PC12 cell when ALK5 or PI3K were inhibited. (**A**) Proliferation of PC12 cells treated with GDF11 (100 ng/mL), SB431542 (20 μM) or LY294002 (1 μM) was assessed by RTCA. (**B**) SB431542 (20 μM) or LY294002 (1 μM) inhibited GDF11-induced suppression of migration. (**C**) Quantitative analysis of (**B**). (**D**,**E**) were the transcriptional levels of nestin, βIII-tubulin, GFAP, cyclin D2 and cyclin B1, respectively. * *p* < 0.05, ** *p* < 0.01, *** *p* < 0.001.

**Table 1 ijms-23-12279-t001:** Summary of TMT-Based LC-MS/MS analysis.

Total Spectrums	Matched Spectrums	Peptides	Unique Peptides	Identified Proteins	Quantifiable Proteins	DEPs
784,796	79,915	18,504	17,878	5452	5209	165

## Data Availability

The data that support the findings of this study are available from the corresponding author upon reasonable request.

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
