# Peer review of "GDF11 Regulates PC12 Neural Stem Cells via ALK5-Dependent PI3K-Akt Signaling Pathway"

_ijms, 2022, doi:10.3390/ijms232012279_

Round 1
Reviewer 1 Report
Scientifically interesting and high-quality manuscript. I have no questions.
Author Response
We thank the reviewer for the positive comments.
Reviewer 2 Report
You will find below some suggestions for small corrections and some issues that may deserve some reflection.
Line 46: “…with the features of long disease duration, slow progress and complex etiology, remains incurable, and even there are few drugs can prevent or delay its progress.” Should be: “…with the features of long disease duration, slow progress and complex etiology, remains incurable, and even there are few drugs that can prevent or delay its progress.” Please correct.
Line 56: “In 2013, Loffredo et al. 5 identified…”: Loffredo et al is not reference 5. Please check all references cited both in the text and in the reference list because more errors can be identified, making it very complicated to identify the content cited in the respective references.
Line 72: in this last paragraph of the introduction, it would be expected that the objective(s) of the work carried out was presented (similarly to what we can find in the abstract) but it is a description of what was done and of the results observed.
Line 77: When presenting the methods used to verify the results, it does not seem right that you include “etc.” Either you mention all the methods used, or you just state that the results were verified using various methods that are described in the Methods section (or something equivalent).
In the same paragraph, you refer what was observed in relation to GDF11, but can you conclude, with your results, if GDF11 is a “rejuvenating factor or a pro-ageing factor”, as mentioned before?
Line 82: Shouldn't the times referred to in this paragraph (24, 48, and 72 h) be described in section 4.2? It’s a bit confusing.
Figure 4: Parts of the information are quite difficult to read (legend of images A and B, protein names in C)
Line 190: “There are 165 proteins were differentially expressed…” Should be “Results show that
165 proteins were differentially expressed…” (or similar). Regarding this information, figure 4 A-B can give us an idea of the different expression observed, but it does not allow us to quantify how many proteins were differentially expressed and also does not allow us to easily understand what percentage of the total these proteins represent.
Lines 212, 215: again, the use of “etc.” to present your results does not seem adequate; avoid using it.
The discussion seems to fall a little short of expectations. Perhaps it would be possible to explore a bit more the consequences that can result from the effect of the GDF11 on the different signaling pathways. What can be expected in other cell types or how can we extrapolate the results observed in vitro to what might happen in vivo? What limitations might hinder this extrapolation?
Line 344: “…and culture….” Should be : “…and cultured….”
Line 351: “… different concentrations of GDF11 (12.5, 25, 50 and 100 ng/mL; R&D systems, USA) were added to incubate for further 5-120 h, respectively.” For each concentration, did you incubate for a different period?
Section 4.6: It would be much easier for readers if you presented the antibody designations in a table or if these were in bold to distinguish them from the brands shown.
Section 4.7: You refer that “proteins were reduced, alkylated and digested according to the protocols.” Are you referring to the citations presented in the previous sentence (2, 19)? Are these correct?
Line 449: “…protein with a fold change of > ± 1.2 (p <0.05) was presented as a DEP.” Please define DEP in the first time this acronym is used in the text. Why did you choose a fold change of 1.2 or higher?
Line 459: “A total of 304 protein and phosphorylated sites in 16 signaling pathways were analyzed.” Does the total of 304 refer to both proteins and phosphorylated sites counted simultaneously? Why did they choose to add the two together, if that is the case? Can't a protein have more than one phosphorylation site?
Conclusion: Do you feel that the presented conclusions respond to the objective defined in the abstract?
Reviewer 3 Report
The authors in deep analysed the effects mediated by GDF11 (Growth factor 11) on neural differentiation process using rat PC12 pheocromocytoma cells as neural model by proteomic assay.
The study was interesting, well designed, clearly described; the experimental approach was appropriated ; statistical analysis robust; the conclusions were supported by the results.
Although the study was conducted in deep, using different biological and proteomic approaches, there are minor points to address:
1. In my opinion the title was not appropriate to respect the results. I suggest to change as: GDF11 regulates PC12 neural stem cells differentiation through ALK5-dependent PI3K-Akt signaling pathway ;
2. In the discussion, the authors could stress that other factors like miRNAs could be involved in the neural differentiation process and the possible interaction with ALK5/PI3K pathway.
